# Longitudinal observational (single cohort) study on the causes of trypanocide failure in cases of African animal trypanosomosis in cattle near wildlife protected areas of Northern Tanzania

**Shauna Richards**[1,2]*, **Davide Pagnossin**[2], **Paul Samson Buyugu**[3], **Oliver Manangwa**[4], **Furaha Mramba**[5], **Emmanuel Sindoya**[6], **Edith Paxton**[7], **Steve J. Torr**[8], **Ryan Ritchie**[9], **Giovanni E. Rossi**[10], **Lawrence Nnadozie Anyanwu**[2], **Michael P. Barrett**[9], **Liam J. Morrison**[7], **Harriet Auty**[2]*

1 International Livestock Research Institute, Nairobi, Kenya, 2 School of Biodiversity, One Health and Veterinary Medicine, University of Glasgow, Glasgow, Scotland, United Kingdom, 3 Sokoine University of Agriculture, Morogoro, Tanzania, 4 Vector and Vector-borne Disease Institute, Tanga, Tanzania, 5 Private Researcher and Medical Entomologist, Tanga, Tanzania, 6 Serengeti District Livestock Office, Mugumu, Tanzania, 7 Roslin Institute, Royal (Dick) School of Veterinary Studies, University of Edinburgh, Edinburgh, Scotland, United Kingdom, 8 Liverpool School of Tropical Medicine, Liverpool, England, United Kingdom, 9 School of Infection and Immunity, University of Glasgow, Glasgow, Scotland, United Kingdom, 10 School of Chemistry, University of Glasgow, Glasgow, Scotland, United Kingdom

* s.richards@cgiar.org (SR); harriet.auty@glasgow.ac.uk (HA)

## Abstract

African animal trypanosomosis (AAT) in cattle is primarily managed through trypanocide administration and insecticide application. Trypanocides can be used for both treatment and prophylaxis, but failure is often reported; this may occur due to resistance, substandard drugs, or inappropriate administration. This study in Tanzania aims to quantify reasons for trypanocide failure. An observational year-long longitudinal study was conducted in high-risk AAT areas in Serengeti District between June 2021-October 2022. Purposive sampling targeted herds with high utilization of the prophylactic trypanocide isometamidium chloride (ISM). When a farmer administered a trypanocide (ISM, diminazine aceturate, homidium), the project veterinarian assessed administration and treatment outcomes were determined based on PCR results from blood samples. A multivariable mixed model was utilized to evaluate risk factors for prophylaxis failure. Quality analysis was performed on trypanocide samples using High Performance Liquid Chromatography.

A total of 630 cattle from 21 farms were monitored for a year-long period. A total of 295 trypanocide administrations were reported, predominantly being ISM (56%) used for prophylaxis (87%). One-third of trypanocide administrations were not given adequately, and many trypanocides were given to animals that tested negative for trypanosome infections by PCR. Failures occurred in 7% (95% CI 3.0–14%) of curative treatments, and 44% (95% CI 35–42%) of prophylactic administrations. The brand of ISM was significantly associated with odds of prophylaxis failure (p = 0.011). On quality analysis, two ISM samples had no

**Data Availability Statement:** Data is available at https://github.com/harrietauty/Tryp_treatmentfailure.

**Funding:** This work was funded by the United Kingdom Biotechnology and Biological Sciences Research Council (BBSRC www.ukri.org/councils/bbsrc/), Grant/Award Numbers BB/S000143/2 & BB/S00243X/1 (awarded to LJM, HA, SJT, MPB), and supported by Global Alliance for Veterinary Medicines (GALVmed www.galvmed.org) with funding from Bill & Melinda Gates Foundation (Investment ID OPP1093639) and UKAID (Project 300504). LJM and EP were supported by core funding to the Roslin Institute by BBSRC, Grant/Award Numbers BBS/E/D/20002173 & BBS/E/RL/230002C. Trypanocide quality analyses were supported by funding from a BBSRC Impact Accelerator award BB/X511110/1 to MPB and HA. The funders had no role in study design, data collection and analysis, decision to publish, or preparation of the manuscript.

**Competing interests:** The authors have declared that no competing interests exist.

detectable ISM isomers, but the remainder of ISM and DA samples (n = 46) fell within the range of acceptable levels. Drug counterfeiting, inadequate use of trypanocides, and resistance are all contributing to trypanocide failure, limiting effective AAT control and with implications for human disease risk. In order to curb trypanocide failure a multi-modal approach to managing the use of trypanocides is required to address all contributing factors.

## Author summary

African animal trypanosomosis (AAT) in cattle is commonly controlled through the use of medications called trypanocides. However, these interventions often fail to control AAT due to reasons such as drug resistance, poor-quality medicines, or incorrect administration. This study, conducted in Tanzania between June 2021 and October 2022, aimed to understand why trypanocides fail. Researchers focused particularly on cattle herds that used a preventive drug called isometamidium chloride (ISM).

During the study, a veterinarian monitored how farmers administered trypanocides like ISM, diminazine aceturate (DA), or homidium, ensuring proper dosage and storage. Blood samples from cattle were tested to check if treatments were effective. The study found that one-third of trypanocide treatments were not given correctly, and many cattle treated with trypanocides were not actually infected. It was determined that 7% of treatments and 44% of preventive doses failed. The type of ISM brand used played a significant role in failure rates, with some brands performing worse than others. Additionally, tests revealed that some ISM samples lacked the active ingredients necessary for effectiveness. The study highlights that counterfeit drugs, improper use, and drug resistance all contribute to the problem, and a more comprehensive strategy is needed to tackle these issues.

## Introduction

African animal trypanosomosis (AAT) is an infectious zoonosis of livestock and wild animals prevalent across much of sub-Saharan Africa (SSA), caused by several species of the protozoan parasite *Trypanosoma spp.* and vectored by tsetse flies. Within cattle, *Trypanosoma congolense*, *Trypanosoma vivax* and *Trypanosoma brucei* cause symptomatic infections characterised by persistent chronic duration and impacts on many aspects of productivity, with high levels of mortality in untreated animals, especially if animals are already in poor condition [1]. *T. brucei rhodesiense* and *T. brucei gambiense* cause human African trypanosomiasis (known as rHAT and gHAT respectively). HAT and AAT are often co-endemic, and in some rHAT foci in East Africa, cattle are also reservoir hosts for *T. brucei rhodesiense* [2,3]. Transmission and management of HAT and AAT are therefore closely related. Trypanosomes are characterized by having highly variable surface antigens, which leads to an inability to mount an effective antibody response, and only recently has there been experimental proof of principle to suggest a vaccine may be possible [4]. Trypanocides constitute the mainstay of control of AAT [5,6]. It is estimated that 35–50 million doses of trypanocide are used on a yearly basis in SSA [7–9], and the use of trypanocides is estimated to cost 20–90 million USD per year [10–12]; however, both the estimated usage and costs are limited as they lack robust calculation methods [5]. Trypanocide use is also a key aspect of control outlined in the Progressive Control Pathway for AAT [13].

AAT control in cattle in SSA is mainly through farmer-led efforts, via prophylactic and curative treatment (from here on out referred to as treatment) trypanocides alongside insecticide application of cattle via dipping or spraying to reduce the tsetse vector of AAT [5,6,14–17]. Diminazene aceturate (DA), isometamidium chloride (ISM), and homidium bromide/chloride (HM) are the trypanocides used to treat AAT in SSA, with DA being the most commonly used amongst the limited published reports [5,18–22]. ISM has a particularly long half-life after administration, which means that ISM is also licensed for prophylaxis–if given at the recommended dose, ISM is indicated to prevent reinfection for around three, or potentially up to six months [22].

Treatment failure and reports of resistance of trypanosomes to trypanocides are reported across SSA [5,18,23–27]. Although failures of treatment or prophylaxis are often assumed to reflect resistance, there are several other potential reasons for failure. Farmers commonly make decisions on treatments without veterinary oversight or access to diagnostics [5,28]; for example, in a study in Kenya 54% of treatments with trypanocides were administered to cattle that did not have symptoms consistent with AAT [29]. In cases where cattle do have AAT, treatment failure or adverse drug reactions can occur due to under- and over-dosing of trypanocides, respectively [20,28,30], using expired or poorly stored trypanocides [6,31,32], and using an inappropriate injection route or technique [6,20,28]. Quality of the available trypanocides is also of concern, with reports of tested products not meeting standards 28–52% of the time [19,33,34]. As well as directly causing treatment failure, these factors may predispose to the development of resistance. There is currently no validated marker for resistant trypanosomes in AAT infections, with current diagnostic methods requiring infection trials in mice or cattle [35]. The relative importance of these factors in causing failures of treatment and prophylaxis is not known. Better understanding of the frequency of trypanocide failure and the relative importance of these factors in failure is essential in improving successful trypanocide use, both with existing drugs, and, with potential new drugs on the horizon [36], to guide the use of new trypanocides that might become available in the future. Effective trypanocide use also has impacts for HAT risk; farmer-led control activities implemented on livestock can impact HAT risk [15], and mass administration of trypanocides to cattle has been used to reduce HAT risk in Uganda [37].

This study was initiated with the overarching aim to test the hypothesis that trypanocide failures were due to a combination of factors (resistance, trypanocide quality/counterfeiting, trypanocide administration and storage methods, and inadequate diagnostic techniques) in Serengeti District, Tanzania. This is a high-risk area for AAT [14,15], with reported high use of trypanocides by farmers. Within this area the project was implemented to test this hypothesis by: (1) evaluating the incidence rate of trypanocide use by farmers on cattle at high risk for trypanocide resistance; (2) determining the proportion of cattle positive for AAT at the time of trypanocide administration by farmers; (3) quantifying failures of treatment and prophylaxis; and (4) determining what factors are associated with trypanocide failure. This was achieved by using an observational longitudinal study over a one-year period on cattle in high-risk areas next to Serengeti National Park.

## Methods

### Ethics statement

The study was approved by the University of Glasgow School of Veterinary Medicine ethics committee. Permission to undertake the study in Tanzania was granted by Tanzania Wildlife Research Institute, and the Commission for Science and Technology (permit number 2021-245-NA-2021-041), and Serengeti District Council. Formal consent was acquired from study

participants and witnessed by the project veterinarian. Written consent was used unless a participant was illiterate. In these cases verbal consent was acquired by the participant, and written consent by a literate family member.

## Study location

The study was conducted in Serengeti district of Mara region, Tanzania from June 2021 to October 2022. The district is located on the Eastern part of Mara region and it is one of the seven districts that constitute Mara region. The district climate includes highland, middle and lowland areas with rainfall influenced by agro ecological zones [38]. The mean temperatures in the district vary from 24°C in wet season to 26°C in the dry season [38]. Much of the district is occupied by protected areas (7,000 km$^2$ is occupied by Serengeti National Park, 190km$^2$ by Ikorongo Game Reserve, 68km$^2$ by Grumeti Game Reserve and 2,456km$^2$ is partially-protected open area) with the remaining area of 659km$^2$ used for agriculture, livestock keeping and residence (Fig 1). The study area is composed of highland savannah mainly with

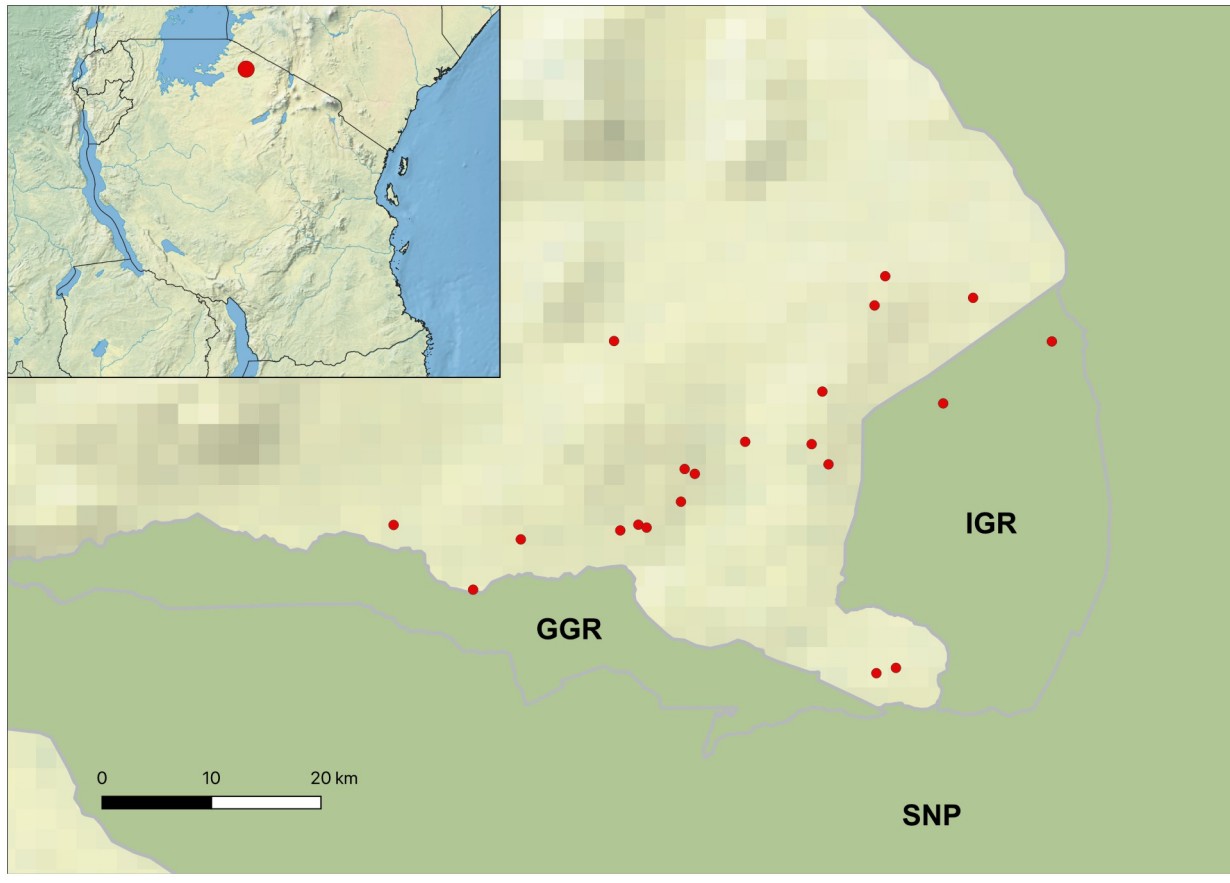

**Fig 1. Geography of the study area.** In the main map, red dots are placed within 2km of the location of each recruited farm (points are jittered to preserve anonymity). SNP: Serengeti National Park; GGR: Grumeti Game Reserve; IGR: Ikorongo Game Reserve. On the top left corner, a single red dot indicates the location of the study area in relation to the whole of Tanzania. Dark green on the map denotes wildlife protected areas [41], whereas light green are non-wildlife protected areas [42]. Most households in this area are livestock keepers and they depend on livestock including cattle, sheep and goats for their income [43]. According to the cattle census for branding [44], the district had 385,542 cattle in 2018. Like other areas in the Northern Tanzania, especially those near national parks and game reserves where tsetse vectors and wildlife hosts are present [45], livestock owners living close to the protected area boundaries are at high risk of AAT, and it is a concern to the members of the community [14].

thorn woodland trees and shrubs in the protected areas, while outside the park is mainly stable agriculture and savannah converted to agriculture, with some areas of remaining savannah habitat [39,40].

## Study participants

Farmers owning cattle were recruited through local livestock officers. Purposive sampling was employed to enrol farmers who were considered at high risk for having cattle affected by trypanocide resistance. In order to be eligible, the farmers and their cattle had to reside in Serengeti District, be located near high-risk areas for tsetse flies (such as wildlife protected areas) and graze cattle near these areas, have 30 or more adult cattle, self-report high use of ISM and other trypanocides, and be willing to participate in the study. Study size was informed by a prior pilot study where 183/772 animals had a trypanocide administered in the last six months. Therefore, over a one-year period monitoring 600 animals we anticipated 280 doses of trypanocides to be administered. This gave us sufficient power to estimate the prevalence of treatment failure (a single proportion), assuming an apparent prevalence of 0.5 (the most liberal assumption) at a precision of 0.06, reflecting a pragmatic balance between field logistics and precision. At enrolment farmers provided informed consent to participate in the study. Farmers were provided with airtime top up vouchers for ~1.90 USD/month to enable them to call the project team and were thanked for their participation with a gift of gumboots and a raincoat. Farmers were called monthly to encourage engagement.

## Study design and sampling

A longitudinal observational study (single cohort) was used to evaluate a range of risk factors leading to trypanocide failure in cases of farmer-administered AAT treatment or prophylaxis. At the study onset a historical questionnaire on farming practices related to cattle health management inclusive of AAT risk factors, and trypanocide use was administered. At recruitment, 30 cattle >6 months of age were randomly selected for inclusion on each farm and identified with an ear tag. Baseline data from cattle was collected by a project veterinarian and field assistant on age, coat colour, pregnancy status, heart girth measurement to estimate weight [46], body condition score (BCS) [47], an abbreviated physical exam was performed on all animals and if the animal appeared unhealthy as determined by the project veterinarian, then a more thorough exam was completed. Details on history of proximity to tsetse and wildlife was collected, as was the last date of insecticide application, last date of trypanocide use for HM, ISM, and DA, and any pharmaceuticals administered in the last 30 days. Blood was sampled from the jugular vein into EDTA and PAXgene (Qiagen) tubes in order to collect samples for packed cell volume (PCV), field trypanosome identification (phase contrast microscopy using the buffy coat method [48]), and DNA extraction for PCR identification of trypanosomes.

Cattle were monitored for 12 months after enrolment. Farmers were asked to alert the project team whenever they planned to administer a trypanocide to their cattle. Whenever possible, the project veterinarian would visit the farm, and the animal would be sampled prior to trypanocide administration, one week, and one month after administration. In the case of ISM, additional samples were collected at 2- and 3-months post trypanocide administration. At these visits the project veterinarian and/or field assistant would perform a physical exam, BCS, heart girth measurement, risk factors for exposure to AAT (wildlife proximity inclusive of grazing practices), and blood collection for PCV, trypanosome identification (microscopy and DNA extraction for PCR). Data were also collected on method and competency of administration of the trypanocide (route of administration), the diluent used, the dose, source, brand and type of trypanocide used, and how the product was stored on farm. This was achieved through

the project veterinarian or field assistant observing the farmer administer the trypanocide and asking questions if required. Blood samples were stored in a cool box to return to the field laboratory; Paxgene tubes were then stored at -20°C until transport and processing.

Any cattle losses during the study period and the causes were recorded. In instances where the loss was not reported by the farmer as it occurred, the loss was determined to be at the mid-point between the notice of loss and the last time the animal was recorded on farm. As cattle could be treated more than one time over the year-long observational period, multiple treatment and follow up periods could occur for each animal enrolled. In the instance when a farmer applied a second dose of trypanocide within a follow up period, the decision on if it constituted a new treatment was as follows: If the second dose was applied within 7 days or less of the first dose then this was collectively considered as one treatment and the same follow up period followed (unless ISM was the second dose and a longer follow up period was required). If the second dose was given more than 7 days after the first dose, then this was considered a new treatment, and a new follow up period was evaluated following this second dose in the series.

Alongside the longitudinal observational study, evaluation of trypanocide quality was also performed. Starting in January 2022, when farmers initiated a treatment, a trypanocide sample was taken to evaluate for quality. On the same day that the treatment was applied, the project veterinarian would visit the site where the farmer had purchased the trypanocide and buy the same product (i.e. matched by brand, producer, dose formulation, lot number, and expiry date). In cases where the same batch was no longer available, a sample of the same product was taken but a different batch number. Samples were stored in sealed plastic bags with a desiccant sachet. In addition, trypanocide samples were collected from local drug shops (formal sector) and livestock markets (informal sector) to assess the quality of trypanocides available in the local area. Five shops and five markets were randomly selected from a list of all shops/markets in the district. Samples of each available trypanocide and brand were purchased from each shop, and up to five samples of different trypanocides and brands from each market.

## Alternations to study following SARS-CoV2 pandemic

Initial recruitment activities began in March 2020, but data collection was delayed due to COVID 19. The study ultimately started in June 2021 with COVID risk management protocols in place and additional recruitment to ensure all participants met the eligibility criteria.

## Diagnostic procedures

Trypanosoma parasitaemia was diagnosed via PCR. DNA was extracted from 8 ml of whole blood samples using the PAXgene Blood DNA Kit (Qiagen) as per manufacturer's instructions. Due to differences in performance for detecting specific trypanosome species [49] and to increase sensitivity, two sets of PCR primers were used, with each performed in triplicate. An internal transcribed spacer (ITS) region-based PCR was used to detect *T. brucei* s.l., *T. congolense*, *T. vivax*, and differentiate from the non-pathogenic *T. theileri* on the basis of different band sizes [50], and species-specific primers were used to target and identify *T. brucei* s.l., *T. congolense* Savannah, and *T. vivax* [51]. PCR conditions used were as previously described. A sample was classed as positive by PCR if there was a visible band of the correct size following gel electrophoresis of PCR products for *T. brucei*, *T. congolense*, or *T. vivax*, on any of the PCR tests or replicates.

## Trypanocide quality testing

Samples returned from Tanzania were assessed for packaging to ascertain deviations from normal.

Powder from each package was carefully weighed into a glass bijou and solutions made up to 29.3 ppm for DA based on the amount of active ingredient presumed to be in the packet according to manufacturer's labelling, or for ISM made up to 60ppm based on an assumption it was made up to at least 60% based on guidelines referenced in Sutcliffe et al. [11].

Sample concentration was quantified using High-Performance Liquid Chromatography (HPLC). The HPLC method used was a 7-minute isocratic 60:40(Acetonitrile 0.1% FA: water 0.1% FA (Fisher Optima LC/Ms Grade) at a flow rate of 0.4ml/min. All measurements were made on a 1260 Agilent Infinity II system using a Phenomenex Gemini C18 150mm column. The injection volume for all measurements was 10ul. Three wavelengths were monitored 380, 320 and 254nm (DAD detector G7115A). The peak area at a wavelength of 320nm was used to calibrate and quantify both DA and ISM. Calibration curves was made using solutions of 5, 10, 20, 30, 40, 60, 80, and 100ppm of both DA (Sigma) and ISM (Veridium, Ceva, Spain)) made up in $H_2O$:Methanol (1:1 + 0.1% formic acid) and were made prior to running a batch of samples. All measurements were made in triplicate and a blank was ran at the start, the end and between each sample in the batch.

## Trypanocide administration outcomes

A logical decision process based on Trypanosoma detection in blood samples by PCR was applied to classify administration outcomes. *Treatment* outcomes were considered for administration of any trypanocide, where the sample at the time of treatment or at the one week follow up tested positive for Trypanosoma, whereas *prophylaxis* outcomes were assessed only in the case of ISM use. When ISM was administered, the animal could be evaluated for both treatment and prophylaxis outcomes. In general, a treatment was considered successful if the animal tested positive for Trypanosoma at the time of treatment and/or one week follow up, then tested negative at one month follow up. A prophylaxis administration was considered successful when it prevented new Trypanosoma infections during the three months that followed an ISM administration. However, decisions on success and failure were complicated. To ensure transparency in classification, a decision tree summarising the outcome classification process was designed to describe plausible patterns of PCR results (Fig 2). A description of the assumptions and limitations associated with each of the outcome pathways is available in S1 and S2 Tables.

## Data management and statistical analyses

All data were collected using Android OS tablets using KoBoCollect and stored in KoBoToolbox. See S1 Box for variable definitions used to interpret data collected in the field.

An animal was considered infected at the time of treatment if laboratory results indicated Trypanosoma presence by PCR detection on the day of treatment or one week later (follow-up 1). Animals testing positive at the one week follow up were included because Trypanosoma parasitaemia levels periodically decrease below the PCR detection threshold [52], and parasite DNA can likely persist for some time, as has been reported for malaria [53]; we assume that parasite DNA may become detectable following treatment due to circulating trypanosome DNA (S1 and S2 Tables).

When classifying trypanocide administrations, animals were considered to show clinical signs consistent with AAT when at least one of the following signs were documented from the

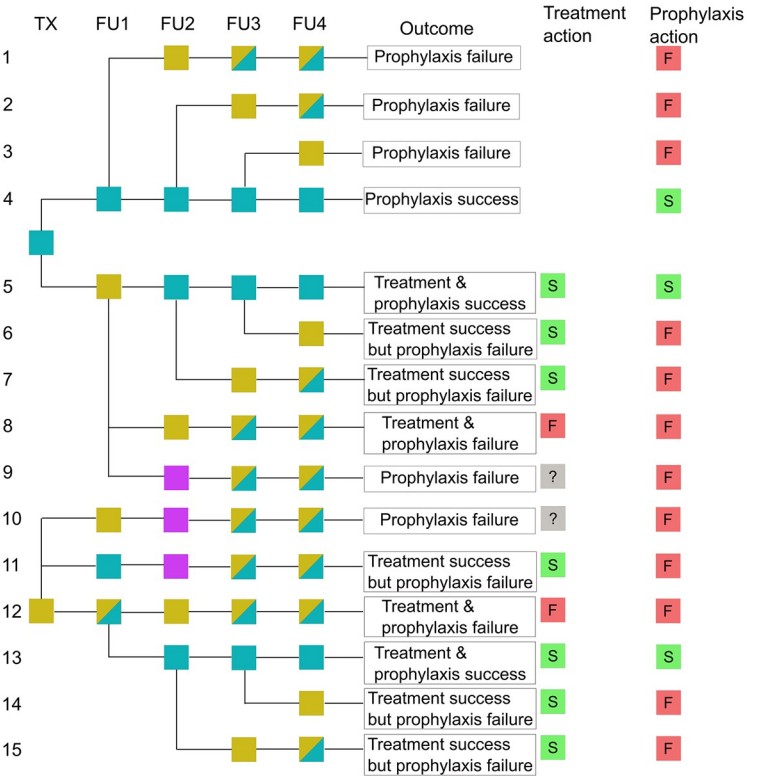

**Fig 2. Decision tree designed to interpret laboratory results and determine trypanocide administration outcomes.**
TX: trypanocide administration day; FU1-4: follow up one-four (one week, one month, two months and three months post treatment). ISM can be use for both treatment and prophylaxis, so both outcomes are considered. DA and HM are used for treatment only.

veterinary examination and PCV assessment: diarrhoea, emaciation, rough hair coat, pale mucous membranes, PCV<25 or BCS < = 1.5. If clinical signs were observed but they were not consistent with AAT, these events were classed as other disease. If no clinical signs were observed, animals were classed as healthy.

The rate of trypanocide administration was calculated by record of new administration for all study animals over the study period (numerator). For time at risk (denominator), the exact time at risk (TAR) was calculated for those cattle who were not lost to follow up (meaning approximately 1 year TAR/cow). If a cow was lost to follow up and the date they exited the herd was known, exact TAR up till the date lost was included, if the date of loss to follow up was not known, it was estimated based on the midpoint between the time the cow was last known/seen in the herd and the date it was identified lost to follow up.

Data analysis was carried out in R Statistical Software (v4.1.1; [54]) using the following packages: dplyr, lubridate, lme4 and car [55–57]. Multivariable analysis of risk factors potentially associated with prophylaxis failure was carried out by constructing a mixed effect logistic regression model. Treatment adequacy, trypanocide brand, sex, age, pregnancy status, farm distance from the closest wildlife protected area (either Serengeti National Park, Grumeti Game Reserve or Ikorongo Game Reserve), herd size and season, were modelled as fixed effects, while herd identification number was modelled as random effect (i.e. to allow for clustering at herd level, as there were limited repeated measures on individual cows). The model was built using a backwards elimination process, applying a likelihood ratio test with a significant p value $< 0.05$.

## Results

A total of 630 cattle were recruited from 21 farms, with herd sizes ranging from 33 to 603 cows. We began staggered enrolment for farms starting in June 2021; data collection ended at latest in October 2022. The majority of participating farmers (n = 18/21) reported recent sightings of tsetse flies and wildlife in close proximity to their herds within the 30 days prior to enrolment (S3 Table). Of the cattle enrolled, 75% (475/630) were female, with an average age of 3.5 years. In contrast, the male population had an average age of 1.7 years (S3 Table).

### Descriptive analysis of trypanocide use practices

Of the 21 farms enrolled, 17 farmers administered trypanocides to at least one of their cattle recruited in the study. We observed farmers administering trypanocides to 38% (242/630) of the enrolled cattle at least once over the study period. Thirty-six animals were lost to follow up during the observation period. A total of 295 trypanocide administrations were reported and documented throughout the study period (S1 Fig). While typically a single trypanocide was administered to each animal, occasionally (n = 16) two trypanocides were simultaneously administered. ISM was the most used trypanocide (n = 174/311, 56% of administrations), followed by DA (n = 117/311, 38%) and HM (n = 20/311, 6%). ISM was primarily used to provide prophylactic protection to animals that farmers viewed as healthy (87% of ISM administrations), whereas DA and HM were administered almost exclusively to animals that farmers perceived to be suffering from AAT (96% and 100% of administrations, respectively). From the 156 trypanocide administrations where farmers indicated they were using trypanocides for prophylaxis, 3% (n = 5/156) had clinical signs consistent with AAT, whilst in 139 administrations where farmers indicated that they were using trypanocides to treat AAT, 27% (n = 37/139) had clinical signs consistent with AAT on veterinary examination (Figs 3 and S2). The rate of trypanocide use in the study cohort was 0.47 administrations/cow-year, 95% CI [0.42, 0.53].

Of the 295 trypanocide administrations recorded, 268 were directly observed by the field veterinarian; adequacy of administration was assessed in these cases (S1 Fig). Approximately 1/3 of the trypanocide administrations observed were considered inadequate (Fig 4). ISM was less likely to be overdosed, but more likely to be underdosed, p <0.001 (Fisher's exact test) (Fig

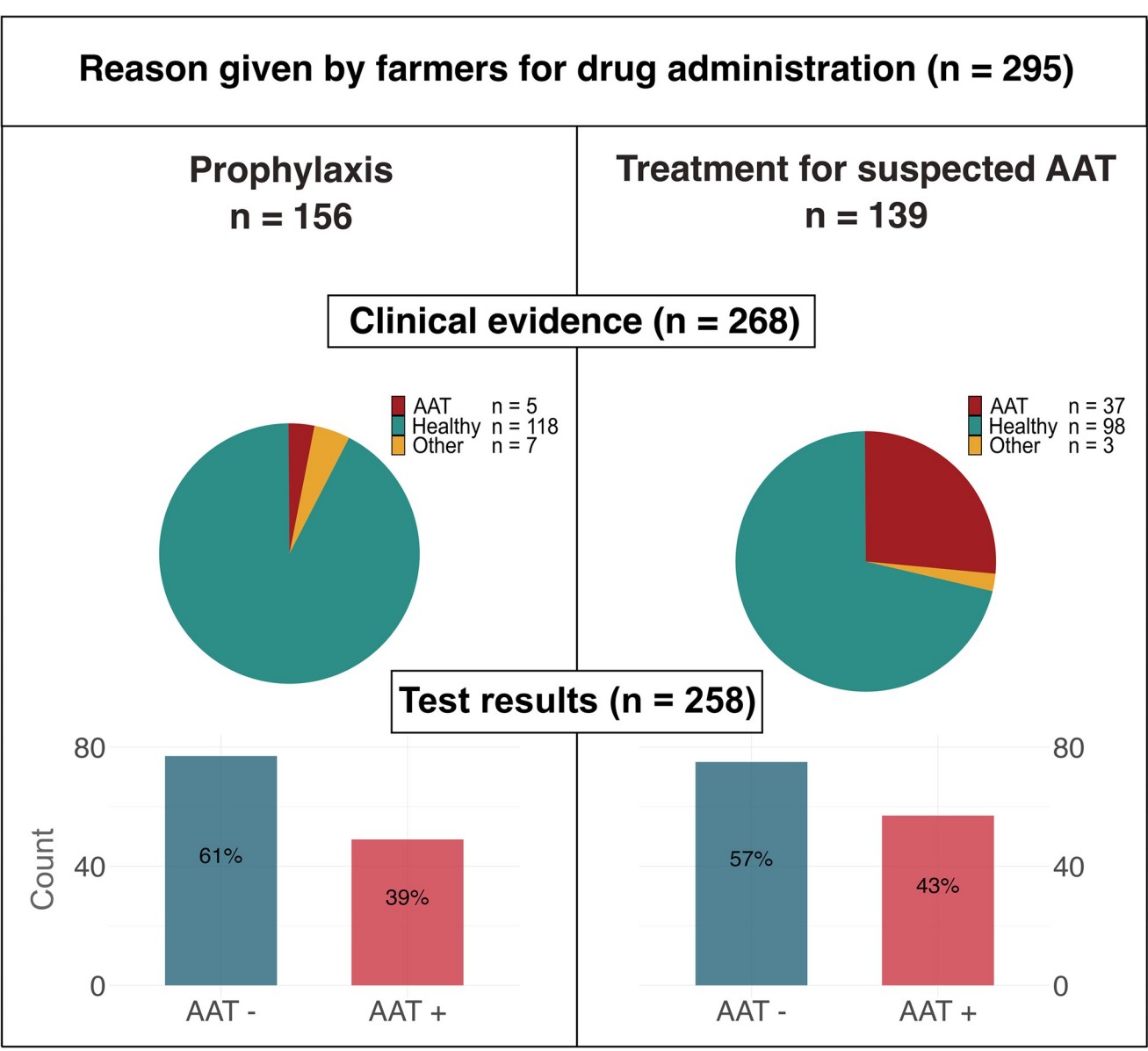

**Fig 3.** Proportion of cases that showed at least one clinical sign consistent with AAT on veterinary examination, and proportion that tested PCR positive (i.e. diagnosis) for at least one of *T. brucei*, *T. congolense* or *T. vivax* at the time of treatment, split between treatment events where farmers indicated they were administering prophylaxis compared to treating animals with AAT *Clinical signs consistent with diarrhoea, emaciation, rough hair coat, pale mucous membranes, PCV<25 or BCS < = 1.5 were classified as AAT. If clinical signs were observed but they were not consistent with AAT, these events were classed as 'other'. If no clinical signs were observed, animals were classed as 'healthy'.

5). Livestock keepers indicated that the mean number of days since insecticide application at the time of trypanocide administration was 10.5 days (range 0–65 days; SD 13.7); all but one farmer reported using synthetic pyrethroid insecticides that are suitable for tsetse control.

### Prevalence of AAT in animals administered trypanocides

Based on the availability of a sample collected immediately prior to treatment and a week later, the prevalence of AAT at the time of treatment was assessed for 258 animals (S1 Fig). The

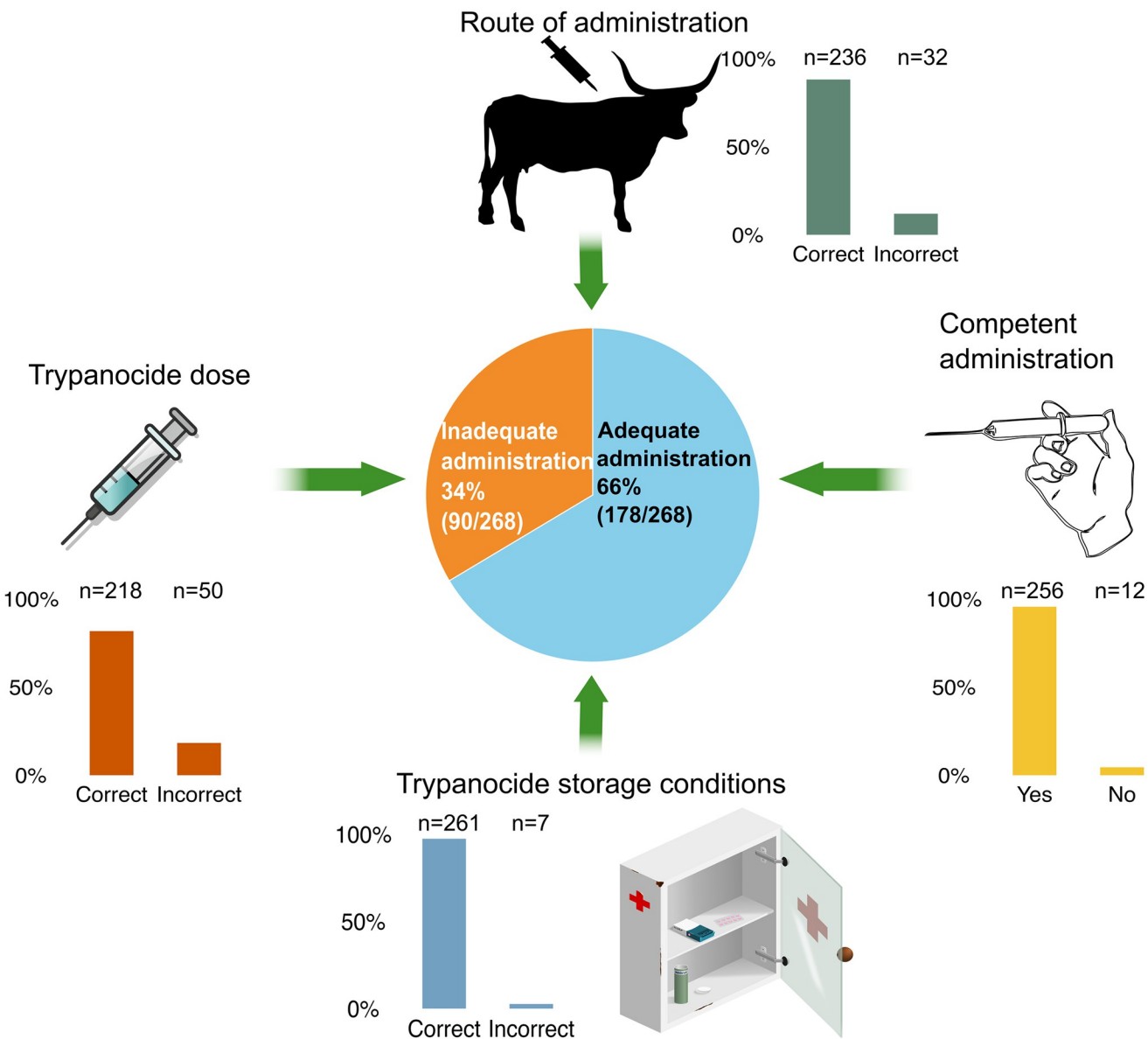

**Fig 4. Proportion of adequate trypanocide administrations out of 268 observed administration events.** Adequacy was established based on four criteria, namely trypanocide dose, route of administration, competency of administration and drug storage conditions.

overall prevalence of AAT (presence of at least one of *T. congolense*, *T. vivax* and *T. brucei*) was 41% at the time of trypanocide administration (n = 106/258). Of the assessed trypanocide administrations, 126 were given a prophylactic drug administration, whereas 132 were treated because of suspected Trypanosoma infection. Of the prophylactic administrations, 39% (n = 49/126) were given to animals already infected. Conversely, only 43% (n = 57/132) of the animals treated for suspected AAT were found positive for Trypanosoma infection (Fig 3). A chi-square test with Yates' continuity correction to compare AAT positivity rates between animals administered a drug for prophylactic purposes and those treated for suspected Trypanosoma infection indicated no statistically significant difference between the positivity rates in the two groups (p = 0.566). *T. congolense* was the species most commonly detected, with a

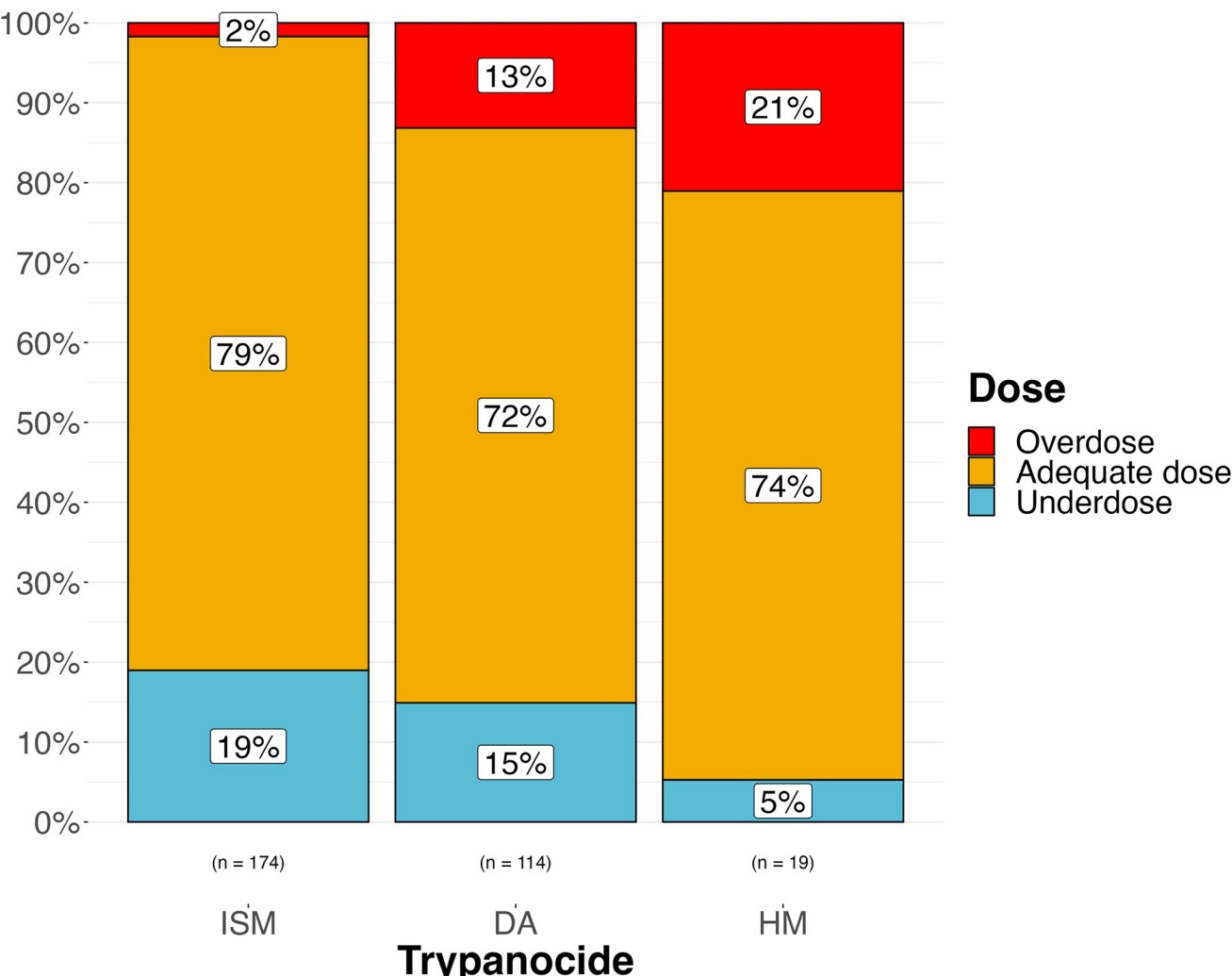

**Fig 5. Proportion of adequate doses, overdoses and underdoses for each trypanocide administered in this study.** ISM = isometamidium chloride; DA = diminazene aceturate; HM = homidium chloride. For ISM, dose categories are based on recommendations to achieve prophylaxis protection (0.5–1 mg/kg). Where multiple trypanocides were combined in one administration, each drug is included separately. The p-value of a Fisher's exact test to compare the association between trypanocide and dose type is reported at p<0.001.

prevalence of 19% in samples at the time of trypanocide administration (n = 50/266), or 9% from all samples inclusive of follow ups (n = 97/1034). The prevalence of *T. brucei* was 7% at the time of trypanocide administration (n = 18/266) and 7% overall (n = 70/1034), while *T. vivax* had a prevalence of 9% at the time of trypanocide administration (n = 25/266) and 6% overall (n = 62/1034). Detailed species-specific prevalence at each time point, are provided in S4 Table.

### Trypanocide administration outcomes

It was possible to determine 257 treatment and 142 prophylaxis pathways (S1 Fig and S1 Table). Many of the treatment pathways (59%, n = 152/257) derived from animals that did not test positive for trypanosome infections by PCR at the time of treatment or at the first follow up, and thus an outcome could not be identified. As for the remaining treatment outcomes,

91% (n = 96/105) were deemed successful, while 7% (n = 7/105) were categorized as failures, and 2% (n = 2/105) remained uncertain (Fig 6). Three of the seven treatment failures were linked to the administration of ISM, three with the use of DA and one with the use of HM. Inadequate administrations, resulting from either a drug underdose or an incorrect route of administration, may have caused three treatment failures. In contrast, four treatment failures

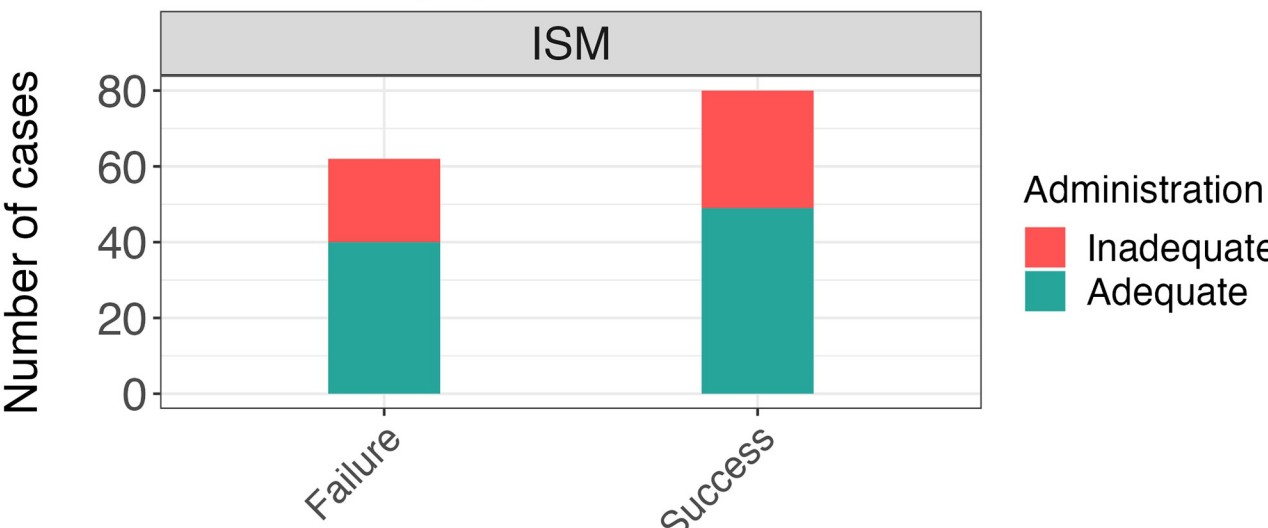

**Fig 6. Number of treatment and prophylaxis outcomes grouped by trypanocide and coloured according to the adequacy of administration.** Treatment outcomes for cattle not infected with Trypanosoma at the time of treatment are not reported. ISM = isometamidium choride; DA = diminazene aceturate; HM = homidium chloride.

occurred despite adequate administrations, suggesting a potential association with unidentified risk factors, drug quality, or drug resistance. Notably, 36% (n = 35/96) of the successful treatments were associated with inadequate administrations.

The prophylactic effect of ISM administrations was considered over the three months following trypanocide use. Among the confirmed prophylaxis outcomes, 56% (n = 80/142) were categorised as successful, whereas 44% (n = 62/142) were considered failures. Of the administrations resulting in prophylaxis failure, 35% (n = 22/62) were associated with inadequate administration. Seventy-one percent of prophylaxis failures occurred within two months of administration of ISM (44/62).

A logistic regression model was built to explore associations of potential risk factors with the failure of ISM prophylactic effect. Of the eight variables identified as potential risk factors, namely treatment adequacy, trypanocide brand, sex, age, pregnancy status, farm distance from the closest high-density wildlife area, herd size and season, only trypanocide manufacturing brand was found to be significantly associated with prophylaxis failure (p = 0.011). Three ISM brands were found to be used within the study population: "Brand A"; "Brand B" both produced in Europe; and "Brand C" produced in Asia. Animals treated with Brand B had 2.44 times higher odds of experiencing a prophylaxis failure than animals treated with Brand A (p < 0.05, 95% CI [1.16, 5.10]), and animals treated with Brand C had 5.48 times higher odds of experiencing prophylaxis failure than animals treated with Brand A (p < 0.05, 95% CI [1.47, 20.4]), (Table 1).

## Drug quality analysis

In total 27 DA samples from six brands (brands DA-A to F) and 21 ISM samples from three brands (brands ISM-A,B,C as described above) were evaluated (S5 and S6 Tables). Twenty-five of the samples were matched to trypanocides administered during the study. All of these samples were compliant with quality testing, with concentrations falling within 15% of expected values (Fig 7). Twenty-two samples were collected via sampling of drug shops and markets. Two ISM samples of brand B had no detectable ISM on HPLC, confirmed by mass spectrometry (Fig 7). The remaining DA and ISM samples were consistent with expected concentrations (Fig 7). Based on the packaging, which notably differed from the expected packaging for this brand (slightly different colour, font less well aligned), the two samples that had no detectable ISM were considered likely to be counterfeit products. These trypanocides had been purchased at livestock markets. Packaging of all other non-tested trypanocides administered during the study was also checked for inconsistencies; one further potential example was identified, also of ISM—brand B, which differed slightly in spelling, colour and alignment. One farmer had administered to this product to 15 cattle; prophylactic failure occurred in ten of the cattle.

## Discussion

The lack of data on the reasons why trypanocide treatment or prophylaxis might fail is limiting the development of effective strategies to make trypanocide use effective and sustainable. This

**Table 1. Summary of the values obtained from the regression model evaluating risk factors for prophylaxis failure following administration of isometamidium chloride.**

| Variable | Reference | Category | Odds ratio | 95% CI | p | $\sigma^2$ | $\tau_{00}$ FarmID |
|---|---|---|---|---|---|---|---|
| Trypanocide brand | Brand A (n = 55) | Brand B (n = 74) | 2.44 | 1.16–5.10 | 0.018 | 3.29 | 1.479142e-14 |
| | | Brand C (n = 13) | 5.48 | 1.47–20.4 | 0.011 | | |

See S5 Table for failure and success incidence by brand

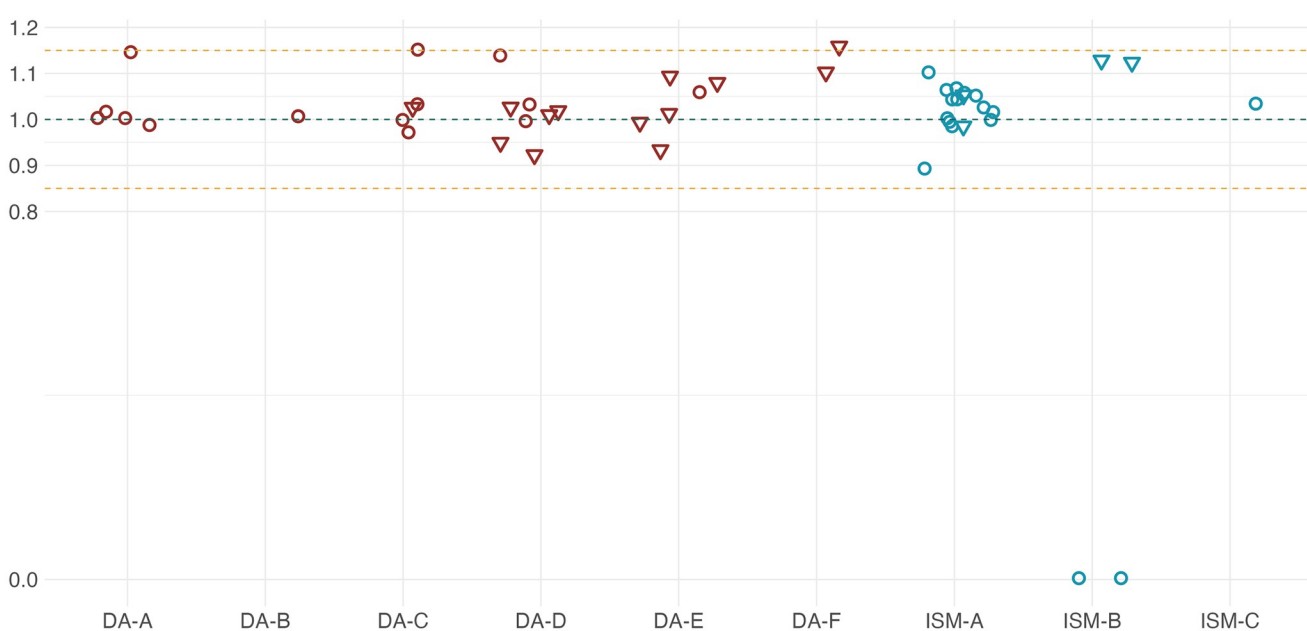

**Fig 7. Ratio between measured and expected drug concentrations in the trypanocide samples analysed by HPLC.** Red shapes indicate samples of diminazene aceturate (DA), while blue shapes represent samples of isometamidium chloride (ISM). Triangles denote samples sourced from the same batches used in trypanocide administrations observed in this study whist circles indicate samples purchased from drug shops and markets. Anonymized labels on the X-axis correspond to different trypanocide brands. The green dashed line represents a perfect match between measured and expected drug concentrations, while the orange dashed lines indicate a deviation of ±15% from the ideal ratio of 1.

study used a unique intensive clinical monitoring approach to assess trypanocide administration and outcomes over a year long period in Serengeti District, Tanzania.

## Incidence rate of Trypanocide use

In a high-risk population for AAT, selected based on high ISM use, the observed incidence is approximately one trypanocide dose per animal every two years. This estimate is conservative for the studied population, as all cattle and farms were included despite suspected under-reporting on four farms. This rate of administration is lower than what was expected in this region given farmer reports prior to the study onset, and reported in the literature [14,15]. This may be in part due to frequent insecticide use, and altered use of trypanocides during the COVID-19 pandemic. ISM became harder to acquire and more expensive during the pandemic, leading to farmers reducing their use of this trypanocide.

Given an estimated 385,542 cattle in this district [44], the number of trypanocide doses administered in a year could be as high as 192,000 doses at a cost of 128,640 USD/year (cost estimate average of actual doses administered 0.67 USD/dose) if farmers all administer trypanocides at this rate using an average drug cost. This estimate is biased by our study population (high use of trypanocides, especially ISM, and located in areas at high risk of AAT) so is unlikely to be representative of the whole district, but it does provide insight into the scale of trypanocide use within high-risk areas. These drug dose cost estimates are based on the mean doses actually administered by farmers and not manufacturer dosing guidelines which would give higher estimates but are provided to understand the real burden of AAT control on live-stock keepers where trypanocide administration is the mainstay of control.

The main trypanocide administered was ISM. This is due in part to our inclusion criteria where we preferentially selected for farms administering ISM; previous surveys in the area have found that DA is most commonly used (S7 Table). Despite its use being contraindicated [11], farmers are still using HM despite its potential toxic and mutagenic effects [22].

## Proportion of AAT-positive cattle at time of trypanocide administration by livestock keepers

In this study, only 43% of cattle that livestock keepers were treating because they perceived that they had AAT, tested positive for *T. congolense*, *T. vivax* or *T. brucei* by PCR. Some of these cases could represent false negatives (i.e. cases where PCR did not detect trypanosomes, but infection was present); measures were taken to enhance sensitivity of PCR, with both ITS and satellite DNA primers used and all PCRs conducted in triplicate, but the fluctuating parasitaemia characteristic of trypanosome infections means that infections will not always be detected, particularly after the initial acute infection [52]. However, PCR sensitivity is very unlikely to fully explain the low proportion testing positive. It is likely that this is indicative of challenges that farmer face in diagnosing AAT in cattle based on clinical signs alone. Symptoms for AAT are not dissimilar to other infectious diseases [58] and these results align with findings in Kenya where more than 50% of trypanocide treatments were given to livestock that did not have clinical symptoms consistent with AAT [29].

The positivity rate by PCR for trypanosomosis for cattle perceived as healthy (by farmers) and receiving prophylactic administration was 39%. This was not significantly different from the positivity rate in cattle that farmers suspected as being unhealthy (p = 0.566). This finding may relate to the stage of infection (i.e. clinical signs are not always consistently detected in chronic cases), and virulence can also vary markedly depending on species and strain of trypanosome [59,60] with some strains causing relatively mild clinical symptoms. Again, the most likely explanation is that accurate diagnosis is difficult when based solely on clinical signs. Diagnostics which are lower in cost than a trypanocide dose would be the ideal solution to allow appropriate administration of trypanocides. Unfortunately, the diagnostics currently available are either too costly, require equipment not readily available in a field setting, or are very insensitive [5].

Farmers were observed in their decision-making processes, and in some cases this was described in detail to the project veterinarian. Farmers may make trypanocide administration decisions based on any obvious clinical symptoms they observe (even those not related to AAT). Although we classified cases as treatment of a sick animal or prophylactic use in a healthy animal based on the farmer's indication, in reality the boundaries are less clear, and farmers are often trying to achieve both, or selecting animals that are doing less well than others for prophylaxis, even if they do not have obvious clinical signs of AAT. Aside from symptoms, farmers also make decisions on trypanocide administrations based on how highly they prioritize cattle within their herd. For example, highly valued cattle may be treated for suspected AAT without any symptoms just in case they have AAT (i.e. reflecting risk mitigation rather than the approach of treating a sick animal). It was also clear within the study population that farmers are often not aware of the differences between trypanocides regarding chemotherapy and chemoprophylaxis and hence do not use them optimally.

## Quantification and factors related to trypanocide failure in treatment and prophylaxis

In our study population, seven percent of treatments and 44% of prophylaxis failed, based on PCR detection of trypanosomes. Three factors were identified as factors contributing to trypanocide failure: i) inadequate administration, ii) drug quality, iii) drug resistance (which was

inferred when failure occurred despite appropriate administration of drugs of sufficient quality).

### Inadequate administration

The study found that around one third of trypanocide administrations were not given adequately. Farmers tended to underdose ISM and overdose DA and HM (Fig 5). This is likely due to ISM being more costly, and being sold in multi-dose sachets, in contrast to DA and HM which are sold in single dose formats. Farmers were often observed to reconstitute ISM multi-dose sachets meant for 8 cattle and administer to 10. With the lower cost of DA and HM, farmers often would not consider dose as carefully and administer the whole dose regardless of the weight of the animal, so smaller or underweight animals could more easily be overdosed whereas very large animals would be underdosed.

Underdosing was the most common (19%) reason for inadequate trypanocide administration (Fig 4), whereas most farmers demonstrated competence in injection technique (96%) and route of administration (88%). One brand (DA-F) of DA was especially prone to underdosing issues (S3 Fig); it is a combination product that had substantially different dilution and dosing requirements compared to all of the other products on the market. This level of competence may not be representative of the general livestock keeping population as our eligibility criteria biased the participants towards more frequent trypanocide usage and increased engagement with veterinary services.

Although the drivers of resistance are not well understood, the continued use of trypanocides at inappropriate doses is likely to increase the risk of resistance developing [8]. It is therefore recommended that farmers be assisted in correctly dosing, administering, and storing trypanocides, as well as when to use them based on clinical signs. It is especially noteworthy that only 43% of cattle that farmers perceived to be infected by AAT did indeed have trypanosome infections detectable by PCR, and only 27% had any clinical signs. This suggests that their perceived burden of infection is higher than the true incidence of AAT cases. Provision of veterinary services is a challenge in more remote rural areas; the Tanzanian government estimates that only 20% of livestock keepers are able to access veterinary services [61]. The reasons for weak animal health systems are complex and difficult to resolve [62]. Digital tools to provide support in decision-making based on clinical signs have shown promise in field trials [63] and could help support more accurate diagnoses and avoid the cost of providing unnecessary treatments.

In cases that tested positive at the time of treatment (n = 105), AAT infection was cleared following treatment, based on whether trypanosomes could be detected by PCR, in 91% of cases (Figs 3 and 6). With only seven cases of treatment failure, no model could be built to evaluate risk factors for treatment failure. These seven cases of treatment failure included cases treated with each of the three trypanocides; in three cases inadequate administration occurred which may have led to those failures (dose, route of administration). The other four cases that did not resolve could have been due to trypanocide resistance; DA resistance has been confirmed in several countries [24,26,27,64]. Interestingly, many successfully resolved cases of AAT were also classified as having inadequate administration (36%). In almost all instances of treatments, cattle rarely had symptoms of AAT, and therefore if they were acute infections without other complicating risk factors, the sub-optimal treatment administered may still be sufficient to lead to cure. Although DA is recommended to be given intramuscularly in cattle, intravenous administration is not uncommon; the efficacy of this route is not clear.

When ISM was administered, 44% of cattle acquired AAT during the three-month period for which ISM is meant to prevent infection with trypanosomes [22]. Since we cannot be sure

that cattle in the study were exposed to AAT during this period, this may be an underestimate of the actual rate of prophylaxis failure. Around one third of the failures occurred in cases where the incorrect route was used, the dose was insufficient, or the trypanocide was not given competently. Notably, some animals treated with ISM intravenously did not exhibit evidence of AAT infection in the three months following administration (although we cannot be certain that they were exposed to AAT). Intravenously administered ISM is rapidly metabolized and not expected to persist in the animal's system to provide prophylaxis, although there is some uncertainty about this [reviewed by 22]. Surprisingly, inadequate treatment was not statistically associated with prophylaxis failure; however, there may be insufficient power to detect this difference. The majority of model variance was at the cow level indicating missing predictors at this level, despite including relevant co-variates.

## Drug quality

Substandard drug quality can arise from both lack of compliance of a manufactured product to align with package concentrations, or from deliberate counterfeit products which may potentially not contain any of the product listed on the package. Within our study, ISM Brand B (European), and Brand C (Asian), were more likely to be associated with prophylaxis failure than Brand A (European). This could reflect differences in production or formulation, or the likelihood of counterfeiting between different brands. Two ISM samples from Brand B were identified as likely counterfeits, with no ISM detectable, and packaging that differed slightly from that expected. In addition, we identified further Brand B samples with suspicious packaging, which were also linked to prophylactic failure. Unfortunately, we did not collect sufficient samples from Brand C to fully test quality in this brand as it was not widely available at the time of trypanocide sampling. Our finding that the odds of prophylactic failure were over five times higher with Brand C is highly suggestive of quality issues, although we cannot completely rule out other unmeasured or confounding factors, such as different animal health practices by the livestock keepers who used this brand.

Substandard quality trypanocides have been identified before in Tanzania [65] as in several other countries [19,33,34], although this is the first study to link them to failure. Substandard quality has been found to be a particular problem in the informal or illegal sector [33] and products that pertain to be produced by Asian brands [19,33]. In this study area, Brand B, officially produced by a European brand, appears to be targeted by counterfeit producers. These findings emphasize the importance of market regulation and enforcement in strengthening the formal markets. However, in some areas livestock keepers struggle to access trypanocides and other veterinary drugs through the formal sector [66], so care needs to be taken to also improve access. The HPLC approach applied to the trypanocide samples gave good accuracy on creating standard curves and only those samples we identified as likely counterfeits fell outside the 15% error limit. We note that the HPLC methodology employed is not capable of distinguishing the relative proportion of each of the four known isomers that are generally co-produced in ISM manufacture. However, all isomers have equivalent trypanocidal activity, and therefore the measurements of the total quantity are a good indication of the quantity of active trypanocide The individual components of commercial ISM do not possess stronger trypanocidal activity than the mixture, nor bypass ISM resistance [67].

DA samples tested were all compliant with expected quantities of trypanocide. DA is lower cost and has simpler manufacturing practices, which may make counterfeiting and sub-standard product less likely. However other studies have reported DA compliance failures [19,33,34], sometimes at higher rates than ISM [33]. Overall the compliance rate was high in our study compared to previous reports of 28–52% [19,33,34].

## Drug resistance

Although this study was not designed to confirm resistance explicitly, treatment or prophylactic failures that occur despite trypanocides of known quality being administered appropriately, are indicative of resistance. In the case of treatments, the four treatment failures that were not associated with inadequate administration could be due to resistance or drug quality.

Approximately one third of prophylactic failures (n = 22/62, 35%) were seen with inadequate administrations, which most often were due to underdosing or incorrect route of administration (S4 Fig). However, 65% were given adequately. Although we did not evaluate drug quality for every batch of ISM used, adequate quality was confirmed in the ISM batches used in 19/40 of these cases, which indicates resistance as the remaining plausible cause for 31% of ISM prophylactic failures (n = 19/62) (S4 Fig). The prophylactic period observed was closer to two months considering both successful and unsuccessful prophylactic trypanocide administrations observed as compared to the reported prophylactic period of 2–3 months and up to 6 months [22]. Confirming resistance relies on infection studies in mice or block treatment approaches in cattle. The resources required to carry out such studies mean they are not commonly performed resulting in a lack of data on quantification of, or risk factors for, resistance. There is an urgent need for identification of markers suitable for identifying resistance.

## Limitations

This is a complex study completed in a field setting with many uncontrolled parameters; however, this setting represents the reality of many livestock keepers dealing with AAT in much of SSA. The outcome rubric developed to understand the possible outcomes of a trypanocide in both treatment and prophylaxis uses, provides a logical and transparent approach that may be of value to other studies (Fig 2). This rubric is built on assumptions (S1 and S2 Tables) that show the evaluation of outcomes of trypanocide administration in a field setting are complex. While PCR is the gold standard to diagnose AAT, it still may lack the sensitivity to detect infections during low parasitaemic stages of AAT infections [52]. Although we can evaluate many of the causes of trypanocide failure, it is still challenging to reach definitive conclusions, and particularly to confirm resistance due to lack of diagnostics available [5].

Although not the aim of this study, it is of note that COVID-19 had a significant impact on trypanocide usage patterns; our data describe the practices post-pandemic. ISM was unavailable June-August 2020 in Serengeti District, and when it returned to the market the price had increased 43–67% depending on the brand. The brand of ISM that became more readily available was at the lower end of the price range and was not well trusted in the study area making it less desirable to farmers. This led to fewer doses of ISM being administered. Additionally, prices of all livestock pharmaceuticals, including insecticides, increased 10–40%, with smaller package sizes having the highest mark up. The overall cost of livestock products could have also impacted farmer decision making through fewer doses used than pre-pandemic, change in brands, and change in prioritization of cattle to receive trypanocides. This highlights the impact that COVID-19 had on animal health practices more widely, and the vulnerability of animal health product markets to global events.

## Conclusions

This study identified and quantified treatment failures in cattle following farmer-led trypanocide administration in Tanzania. Uniquely, this enabled us to assess all the potential causes and confirmed that inadequate administration and drug quality were linked to failure, with remaining failures likely to be caused by resistance. We documented high rates of failure (44%) with ISM in particular. In addition, much of the chemotherapeutic administration was

to cattle where no trypanosome parasitaemia could be detected; it is clear that misdiagnosis and unnecessary treatments are common. In turn, the misuse of trypanocides (dose, site/competency of injection, storage), alongside counterfeit or poor quality trypanocides, can lead to a higher selection pressure for trypanocide resistance and a higher cost to farmers. Candidate drugs are being explored, but if new trypanocides are brought onto the market without consideration for current use practices, these new products will face the same pressures, and their value may not be realised. In addition, since cattle can carry the trypanosomes that cause HAT, ineffective chemotherapy and chemoprophylaxis threaten the potential for livestock disease control to reduce human disease risk.

## Supporting information

**S1 Box. Variable definitions of data collected using KoBoCollect.**
(DOCX)

**S1 Table. Assumptions and limitations of ISM outcome pathways.**
(DOCX)

**S2 Table. Assumptions and limitations of DA and HM outcome pathways.**
(DOCX)

**S3 Table. Summary of baseline data collected during farms and cattle enrolment in the study.**
(DOCX)

**S4 Table. Frequency of detection of *T. brucei*, *T. congolense* and *T. vivax* in cattle blood samples at different time points.** Samples positive for more than one Trypanosoma species are reported as positive in each relevant column.
(DOCX)

**S5 Table. Summary of the use of trypanocide brands administered to the cattle enrolled in this study.**
(DOCX)

**S6 Table. Summary of the distribution of hypothesised risk factors for prophylaxis failure in the study dataset.**
(DOCX)

**S7 Table. Number and proportion of cattle that livestock keepers reported had been treated with diminazene, isometamidium or homidium within the preceding six months in Serengeti District, Tanzania in 2016.**
(DOCX)

**S1 Fig. Flowchart detailing the steps involved in the sample collection and testing process.** Black text summarises action outcomes, whilst red text provides information on data or sample loss. Abbreviations: TX = day of treatment; FU1 = follow-up one; FU2 = follow-up two; FU3 = follow-up three; FU4 = follow-up four; ISM = isometamidium chloride; DA = diminazine aceturate; HM = homidium chloride. Thirty-six samples collected at FU2 from animals treated with DA or HM were not PCR tested because laboratory results from TX and FU1 samples were sufficient to determine treatment outcome.
(PNG)

**S2 Fig. Density plot of packed cell volume (PCV) values from blood samples of animals testing PCR negative and positive for *Trypanosoma congolense*, *T. brucei*, and/or *T. vivax*.**

Values derived from samples collected before trypanocide administration are plotted.
(PNG)

**S3 Fig. Dosage of trypanocide administrations administered by farmers by agent.** Abbreviations: ISM = isometamidium chloride; DA = diminazine aceturate; HM = homidium chloride. Blue squares represent recommended curative dosages for each trypanocide, while an orange square in the ISM column indicates the recommended prophylactic dosage range. Data points are color-coded by brand. Circular data points indicate a trypanocide administered on its own, while triangles denote a trypanocide given in combination with another one.
(PNG)

**S4 Fig. Flowchart describing the classification process of ISM administrations resulting in prophylaxis failure.** After excluding inadequate administrations (n = 22), untested samples (n = 21), and samples of poor drug quality (n = 0), it is plausible to conclude that some prophylaxis failures may have been due to *Trypanosoma* strains resistant to ISM prophylaxis.
(PNG)

## Acknowledgments

We are grateful to Tanzania Wildlife Research Institute (TAWIRI), Tanzania Commission for Science and Technology (COSTECH), Tanzania Ministry of Livestock and Fisheries, and Serengeti District Council for permitting this study to be conducted. We would particularly like to thank the participating livestock keepers for their engagement and patience.

The findings and conclusions contained within are those of the authors and do not necessarily reflect positions or policies of the Bill & Melinda Gates Foundation or the UK Government.

## Author Contributions

**Conceptualization:** Shauna Richards, Oliver Manangwa, Furaha Mramba, Steve J. Torr, Michael P. Barrett, Liam J. Morrison, Harriet Auty.

**Data curation:** Shauna Richards, Davide Pagnossin, Lawrence Nnadozie Anyanwu.

**Formal analysis:** Davide Pagnossin.

**Funding acquisition:** Oliver Manangwa, Furaha Mramba, Steve J. Torr, Michael P. Barrett, Liam J. Morrison, Harriet Auty.

**Investigation:** Shauna Richards, Davide Pagnossin, Paul Samson Buyugu, Emmanuel Sindoya, Edith Paxton, Ryan Ritchie, Giovanni E. Rossi.

**Methodology:** Shauna Richards, Oliver Manangwa, Furaha Mramba, Steve J. Torr, Michael P. Barrett, Liam J. Morrison, Harriet Auty.

**Project administration:** Shauna Richards, Davide Pagnossin, Paul Samson Buyugu, Emmanuel Sindoya, Harriet Auty.

**Resources:** Shauna Richards, Paul Samson Buyugu, Edith Paxton, Ryan Ritchie, Giovanni E. Rossi, Harriet Auty.

**Software:** Shauna Richards, Davide Pagnossin.

**Supervision:** Shauna Richards, Davide Pagnossin, Michael P. Barrett, Liam J. Morrison, Harriet Auty.

**Validation:** Edith Paxton, Ryan Ritchie, Giovanni E. Rossi.

**Visualization:** Davide Pagnossin.

**Writing – original draft:** Shauna Richards, Davide Pagnossin, Paul Samson Buyugu, Harriet Auty.

**Writing – review & editing:** Shauna Richards, Davide Pagnossin, Paul Samson Buyugu, Oliver Manangwa, Furaha Mramba, Emmanuel Sindoya, Edith Paxton, Steve J. Torr, Ryan Ritchie, Giovanni E. Rossi, Lawrence Nnadozie Anyanwu, Michael P. Barrett, Liam J. Morrison, Harriet Auty.

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
