## [Decision Letter · Decision Letter 0]

4 Nov 2024

PNTD-D-24-01315Title Longitudinal Observational (single cohort) Study on the Causes of Trypanocide Failure in cases of African Animal Trypanosomosis in Cattle Near Wildlife Protected Areas of Northern TanzaniaPLOS Neglected Tropical Diseases Dear Dr. Richards, Thank you for submitting your manuscript to PLOS Neglected Tropical Diseases. After careful consideration, we feel that it has merit but does not fully meet PLOS Neglected Tropical Diseases's publication criteria as it currently stands. Therefore, we invite you to submit a revised version of the manuscript that addresses the points raised during the review process. Please submit your revised manuscript within 30 days Jan 03 2025 11:59PM. If you will need more time than this to complete your revisions, please reply to this message or contact the journal office at plosntds@plos.org. Please include the following items when submitting your revised manuscript:*
A rebuttal letter that responds to each point raised by the editor and reviewer(s). You should upload this letter as a separate file labeled 'Response to Reviewers'. This file does not need to include responses to any formatting updates and technical items listed in the 'Journal Requirements' section below.*
A marked-up copy of your manuscript that highlights changes made to the original version. You should upload this as a separate file labeled 'Revised Manuscript with Track Changes'.*
An unmarked version of your revised paper without tracked changes. You should upload this as a separate file labeled 'Manuscript'. If you would like to make changes to your financial disclosure, competing interests statement, or data availability statement, please make these updates within the submission form at the time of resubmission. Guidelines for resubmitting your figure files are available below the reviewer comments at the end of this letter. We look forward to receiving your revised manuscript. Kind regards, Kyoko Hayashida, Ph.D, DVMAcademic EditorPLOS Neglected Tropical Diseases Susan Madison-AntenucciSection EditorPLOS Neglected Tropical Diseases

Shaden Kamhawi

co-Editor-in-Chief

Paul Brindley

co-Editor-in-Chief

 **Journal Requirements:** **Additional Editor Comments (if provided):** In addition to the comments from reviewers, I request you to confirm these two minor points.

1. Check graph and number in "Figure 4 : Trypanocide storage conditions" is correctly shown.

2. Consider result of Fisher's exact test can be explained in the result section, not in the Figure 5.**Reviewers' comments:** Reviewer's Responses to Questions

**Key Review Criteria Required for Acceptance?**

**Methods**

-Are the objectives of the study clearly articulated with a clear testable hypothesis stated?

-Is the study design appropriate to address the stated objectives?

-Is the population clearly described and appropriate for the hypothesis being tested?

-Is the sample size sufficient to ensure adequate power to address the hypothesis being tested?

-Were correct statistical analysis used to support conclusions?

-Are there concerns about ethical or regulatory requirements being met?

Reviewer #1: This is a very interesting study on the use of trypanocides in Tanzania, with the objectives clearly articulated. The study design is adequate and flexible enough to accommodate the challenges posed by the study area and the study requirements. The study population is well characterized in an area of "high risk of trypanocide resistance development" was selected based on a perceived high challenge and perceived intense use of trypanocides. From this area, a total of 630 cattle from 21 farms were monitored for a year-long period. This sample size gave enough power for the majority of the required analyses. The statistical analysis was adequate and sufficient to support the conclusions and the study was conducted following high ethical standards.

Reviewer #2: Lines 280 it should bijou and not bijoux for the bijou bottle

For figure 2 and the decision tree make the font bigger and easier to read

**Results**

-Does the analysis presented match the analysis plan?

-Are the results clearly and completely presented?

-Are the figures (Tables, Images) of sufficient quality for clarity?

Reviewer #1: The results are clearly presented, making good use of a variety of graphic resources.

Reviewer #2: (No Response)

**Conclusions**

-Are the conclusions supported by the data presented?

-Are the limitations of analysis clearly described?

-Do the authors discuss how these data can be helpful to advance our understanding of the topic under study?

-Is public health relevance addressed?

Reviewer #1: The robustness of its experimental design, and chiefly its longitudinal nature, make this study, in a way, a pioneer in the combined study of the complex reasons for failure in the treatment of trypanosome infections, including resistance. This experimental design should be adapted and improved in different ecological niches of sub-Saharan Africa, in order to accurately identify and address the major challenges pertaining the effective use of trypanocides.

The limitations are thoroughly discussed by the authors, as well as the implications of the findings of this work for the treatment of trypanosome infections in Africa. Moreover, the authors established an interesting connection between the failure of the treatment with trypanocides and the reservoir status of cattle for Human African Trypanosomiasis, which points for an important public health dimension.

Reviewer #2: The discussion could be streamlined a bit as they are quite some repetitions between the results and discussion and within the discussion itself. It is quite descriptive and long.

**Editorial and Data Presentation Modifications?**

Reviewer #1: The paper is very clear and well written. I added minor comments directly to the manuscript.

Reviewer #2: (No Response)

**Summary and General Comments**

Reviewer #1: It is a robust, clear and inspiring study.

Reviewer #2: Overall I compliment the authors on this is very interesting piece of work with good methods and interesting results although maybe not as conclusive as one would expect. This is an important and valuable piece of work overall.

PLOS authors have the option to publish the peer review history of their article (what does this mean?). If published, this will include your full peer review and any attached files.

Reviewer #1: No

Reviewer #2: No

---

## [Editor Report · Decision Letter 1]

17 Dec 2024

Dear Dr Richards,

We are pleased to inform you that your manuscript 'Title Longitudinal Observational (single cohort) Study on the Causes of Trypanocide Failure in cases of African Animal Trypanosomosis in Cattle Near Wildlife Protected Areas of Northern Tanzania' has been provisionally accepted for publication in PLOS Neglected Tropical Diseases.

Best regards,

Kyoko Hayashida, Ph.D, DVM

Academic Editor

Susan Madison-Antenucci

Section Editor

Shaden Kamhawi

co-Editor-in-Chief

Paul Brindley

co-Editor-in-Chief

---

## [Editor Report · Acceptance letter]

16 Jan 2025

Dear Dr Richards,

We are delighted to inform you that your manuscript, "Longitudinal Observational (single cohort) Study on the Causes of Trypanocide Failure in cases of African Animal Trypanosomosis in Cattle Near Wildlife Protected Areas of Northern Tanzania," has been formally accepted for publication in PLOS Neglected Tropical Diseases.

Best regards,

Shaden Kamhawi

co-Editor-in-Chief

Paul Brindley

co-Editor-in-Chief
